# Enhancing Stroke Recognition: A Comparative Analysis of Balance and Eyes–Face, Arms, Speech, Time (BE-FAST) and Face, Arms, Speech, Time (FAST) in Identifying Posterior Circulation Strokes

**DOI:** 10.3390/jcm13195912

**Published:** 2024-10-03

**Authors:** Onur Tanglay, Cecilia Cappelen-Smith, Mark W. Parsons, Dennis J. Cordato

**Affiliations:** 1Department of Neurology and Neurophysiology, Liverpool Hospital, Liverpool, NSW 2170, Australia; onur.tanglay@health.nsw.gov.au (O.T.); cecilia.cappelensmith@health.nsw.gov.au (C.C.-S.); mark.parsons@health.nsw.gov.au (M.W.P.); 2South Western Sydney Clinical School, University of New South Wales, Liverpool, NSW 2170, Australia; 3Ingham Institute for Applied Medical Research, Liverpool, NSW 2170, Australia

**Keywords:** stroke, FAST, BE-FAST, posterior circulation

## Abstract

**Background/Objectives**: Posterior circulation stroke (PCS) poses a diagnostic challenge due to the diverse and subtle clinical manifestations. While the FAST (Face, Arms, Speech, Time) mnemonic has proven effective in identifying anterior circulation stroke, its sensitivity to posterior events is less clear. Recently, the addition of Balance and Eyes to the mnemonic has been proposed as a more comprehensive tool for stroke recognition. Despite this, evidence directly comparing the effectiveness of BE-FAST and FAST in identifying PCS remains limited. **Methods**: A retrospective analysis was performed on stroke calls at a comprehensive stroke centre, Sydney, Australia. BE-FAST symptoms first assessed at an emergency department triage were recorded, along with automated acute computerised tomography perfusion (CTP) imaging findings. Haemorrhagic strokes were excluded from analysis. An ischaemic stroke diagnosis was confirmed 48–72 h later with magnetic resonance imaging (MRI) brain. The performance of 1. BE-FAST and FAST and 2. BE-FAST and CTP in the hyperacute detection of posterior circulation ischaemic stroke was compared. **Results**: Out of 164 identified ischaemic infarcts confirmed on MRIs, 46 were PCS. Of these, 27 were FAST-positive, while 45 were BE-FAST-positive. Overall, BE-FAST demonstrated a higher sensitivity compared to FAST in identifying PCS (97.8 vs. 58.7) but suffered from a lower specificity (10.0 vs. 39.8). Notably, 39.1% (*n* = 18) of patients with PCS would have been missed if only FAST were used. Furthermore, of the 26 PCS negative on CTP, 25 were BE-FAST-positive, and 14 were FAST-positive. **Conclusions**: The incorporation of Balance and Eye assessments into the FAST protocol improves PCS detection, although may yield more false positives.

## 1. Introduction

Stroke remains among the leading causes of disability and mortality worldwide [1]. It is the second-leading cause of death and third-leading cause of combined death and disability with 10.3 million new strokes accounting for 113 million disability-adjusted life years (DALYs) per year worldwide [2]. This necessitates timely and accurate recognition for optimal management and outcomes. The last 30 years has seen both a 70% increase in the global burden of incident stroke and major advancements in stroke care with hyperacute therapies such as intravenous thrombolysis (IVT) and endovascular thrombectomy (EVT). Although IVT and EVT demonstrate significant benefits in terms of three-month functional outcomes and reduced stroke-related morbidity, they rely on timely administration [3,4,5,6,7]. Early recognition of stroke is therefore critical in reducing treatment delays and improving patient outcomes. Delays in presentation, diagnosis and misdiagnosis may result in adverse clinical outcomes.

Several prehospital scales have been developed to aid in stroke detection at the prehospital and emergency department (ED) level. These include simple scales such as the three-item Face, Arms, Speech, Time (FAST), the Rapid Arterial Occlusion Evaluation Scale (RACE), the Cincinnati Prehospital Stroke Severity Score (CPSSS), the Los Angeles Prehospital Stroke Screen (LAPSS), the Melbourne Ambulance Stroke Screen (MASS), and the Medic Prehospital Assessment for Code Stroke (Med PACS) [8,9]. In contrast to FAST, which was designed as a simple prehospital stroke detection tool, RACE, which incorporates facial palsy, arm, and leg motor weakness with elements of cortical function (aphasia or agnosia), is an attempt to help health professionals differentiate large vessel occlusion (LVO) from non-LVO [10]. Similarly, CPSSS is a simple tool designed to detect severe strokes and LVO with patients scoring points for reduced level of consciousness to commands and questions, gaze palsy, and arm weakness [11]. Among these stroke scales, FAST was found to have the highest sensitivity at 85% in one cohort [12]. FAST’s straightforward message has also seen it promoted as a public health intervention to improve stroke awareness in the community [13].

In 1995, the National Institute of Neurological Disorders and Stroke (NINDS) stroke trial confirmed the benefit of intravenous tissue plasminogen activator (tPA) in improving three-month outcomes, including a higher proportion of patients with a modified Rankin Scale score of 0–1 (functional independence without disability) when administered within 3 h of ischaemic stroke symptom onset [14]. The time window for thrombolysis benefit in acute ischaemic stroke was later extended to 4.5 h [15]. FAST was a prehospital stroke identification tool originally developed three years after the 1995 NINDS stroke trial, as part of a UK ambulance staff training package, with the aim of expediting the administration of intravenous tPA within 3 h of stroke symptom onset [16]. Its use was found to achieve high levels of detection and diagnostic accuracy of stroke by ambulance paramedics [16].

While simple scales provide the benefit of ease of administration and may require little training, their sensitivity can be limited. One study has shown that the FAST failed to detect 14% of ischaemic strokes, with 70% of FAST-negative patients experiencing either gait balance/leg weakness or visual symptoms [17]. In another study involving patients who presented with posterior circulation ischaemic stroke or transient ischaemic stack (TIA), the percentage of patients who had posterior circulation infarcts who were FAST-negative was 40% [18].

Posterior circulation strokes (PCSs) comprise 20–25% of ischaemic stroke (AIS) presentations [19]. They are difficult to diagnose due to the presence of diverse and often subtle clinical manifestations. Symptoms such as vertigo, ataxia, and visual disturbances may not be adequately captured by traditional stroke scales, which are often focused on the anterior circulation. In one study, up to a third of ED presentations involving posterior circulation infarcts were misdiagnosed as benign disorders [20] and neurological signs were often minor or absent. Non-contrast computerised tomography (CT) brain is typically the first imaging modality performed with a reported sensitivity ranging from 7 to 42% [21]. In addition to demonstrating early infarction, CT evidence of a hyperdense large vessel such as the posterior cerebral or basilar artery can also immediately alert ED staff to the diagnosis of a PCS [22]. CT angiography (CTA) improves sensitivity when compared to a non-contrast CT but may still be normal in the hyperacute setting [20,23]. CT perfusion (CTP) is increasingly becoming the mainstay in quickly assessing AIS. In conjunction with non-contrast CT brain and CTA imaging, it may facilitate the determination that an acute ED presentation is an ischaemic stroke. CTP involves the acquisition of CT brain images following the course of intravenous injection of contrast through the brain to assess tissue characteristics rather than relying solely on clinically time-based criteria [24]. It can measure cerebral blood flow, volume, and the mean transit time of intravenous contrast movement through a brain region affected by stroke. Thresholds have been established which can reliably delineate irreversibly damaged infarction or ‘core’ from hypoperfused brain or ‘penumbra’ [22]. Software algorithms can automatically quantify the amount of mismatch volume between core and penumbra [24]. Its sensitivity in identifying ischaemia of the posterior circulation is lower than that for the anterior circulation. However, CTP-deficit volumes have been found to be an excellent predictor of functional outcome in patients with basilar artery occlusion [25]. Hence, CTP may aid clinicians, including ED staff, in the detection of stroke and its severity.

The shortcomings in diagnosing posterior circulation events and consequent delays in treatment may result in poorer long-term outcomes [26]. This is despite the fact that posterior circulation infarcts still benefit from hyperacute reperfusion therapies [27]. To combat this, the addition of Balance and Eyes to the FAST scale has manifested in BE-FAST, which aims to improve the detection of posterior circulation events. Among public education materials distributed by comprehensive stroke centres in the United States, BE-FAST has become more popular than FAST, suggesting increased uptake [28]. Several studies have compared the two scales in identifying ischaemic strokes, with variable results. A recent meta-analysis demonstrated a higher sensitivity for FAST and a higher specificity and area under the receiver operator characteristic curve (AUC-ROC) for BE-FAST [29]. Despite this, the performance of BE-FAST in the identification of PCS remains unknown.

This paper aims to address this gap by comparing the performance of BE-FAST and FAST in the hyperacute detection of later MRI-confirmed posterior circulation ischaemic strokes among patients presenting to the emergency department (ED), and those in-hospital patients with stroke code alerts. We also sought to compare the utility of automated CTP and BE-FAST in the hyperacute detection of posterior circulation infarcts later confirmed on MRI brain imaging at 48–72 h after presentation.

## 2. Materials and Methods

### 2.1. Data Collection

A retrospective study of consecutive stroke codes activated at the Liverpool Hospital, a tertiary referral and comprehensive stroke centre, in Sydney, Australia, between January 2023 and April 2023 was conducted. These included patients aged 18 years or older who had a stroke code activated by ambulance personnel en route to the ED; stroke codes activated by ED personnel on direct presentations to the ED; and inpatient stroke codes. Haemorrhagic strokes and patients with suspected ischaemic stroke who did not undergo MRI for confirmation were excluded. Patients were also excluded if they were transferred from a peripheral hospital for stroke intervention. Institutional review board approval was obtained prior to the study (SWS Registry 2019/ETH00096).

Patient-reported symptoms during initial assessment were used to determine whether they screened positive for FAST or BE-FAST. The B and E components of BE-FAST were classified as gait imbalance or lower limb weakness, and visual loss or diplopia. Demographics recorded included patient’s age and sex. The ABCD2 and National Institutes of Health Stroke Scale (NIHSS) scores were also recorded, along with local radiologist reports of CT and CTA findings and automated results reported by CTP, and whether stroke intervention (including intravenous thrombolysis and endovascular thrombectomy) was administered. A final diagnosis of ischaemic stroke was determined based on MRI confirmation, most commonly performed between 24 and 72 h after symptom onset, and further stratified into anterior or posterior circulation infarction. The final documented diagnosis for all other cases were also recorded.

### 2.2. Statistical Analysis

Prior to analysis, data were anonymised and coded. The anterior and posterior circulation groups were coded in such a way that the author performing the analysis was blind to the groups during analysis. Demographic data were summarised using median and interquartile range (IQR). Demographic data were compared between anterior and posterior circulation infarcts using the nonparametric Mann–Whitney U test and the Chi-squared test. FAST and BE-FAST were converted into numerical scores ranging from one to three, and one to five, respectively. The sensitivity, specificity, positive predictive value (PPV), and negative predictive value (NPV) of FAST and BE-FAST in detecting AIS were calculated and presented as percentages with 95% confidence intervals, along with separate analyses for anterior and posterior circulation strokes. The performance of automated CTP was analysed in the same way. Receiver operating characteristic curves were plotted for each analysis. Analysis was performed using MedCalc version 22.019 (MedCalc Software Ltd., Ostend, Belgium) and SPSS version 29.0.1 (IBM Corp., Armonk, NY, USA, 2023). Following analysis, data validation was performed independently by three of the authors.

## 3. Results

### 3.1. Patient Characteristics

Of the stroke codes reviewed, 556 patients met the criteria for inclusion. Their demographics are summarised in Table 1. A final diagnosis of AIS was made in 164 (29.5%) patients, while 46 (8.3% of total; 28.0% of AIS) patients had a final diagnosis of posterior circulation infarction. The median NIHSS was higher in anterior circulation strokes (eight vs. four). Among all patients with AIS, intravenous thrombolysis was administered in 22.0% (*n* = 36) and endovascular thrombectomy was performed in 27.4% (*n* = 45). Other diagnoses included TIA in 53 (9.5%) patients, peripheral vestibulopathy in 51 (9.2%) patients, headache including migraine in 45 (8.1%), and delirium in 25 (4.5%) patients.

Face weakness (anterior 48.3%; posterior 23.4%), arm weakness (anterior 71.2%; posterior 44.7%), and speech impairment (anterior 55.9%; posterior 29.8%) were more prevalent in anterior circulation cases compared to posterior circulation cases. In contrast, balance problems or lower limb weakness (anterior 61.0%; posterior 72.3%), and eye signs or visual impairment (anterior 0.8%; posterior 23.4%) were more common in posterior circulation cases.

The presence of acute infarction on non-contrast CT was reported in three (6.5%) of the posterior circulation cases. All three of these cases were CTA- and CTP-positive. The presence of a vessel occlusion in the territory of infarction was reported in eight (17.4%) of the posterior circulation infarcts, with two of these eight cases demonstrating basilar artery occlusion. All eight CTA-positive cases were also CTP-positive.

### 3.2. Performance of BE-FAST in Diagnosing AIS

The performance of BE-FAST and FAST is summarised in Table 2. Among all AIS, BE-FAST was positive in 98.8% of cases, while FAST was positive in 86.6%. The sensitivity and specificity of FAST and BE-FAST for all stroke alerts are shown in Table 2. BE-FAST and FAST had a comparable AUC-ROC of 0.71 (95% CI 0.66–0.75) and 0.70 (0.66–0.75), respectively (Figure 1a).

Of all posterior circulation infarcts, 97.8% were BE-FAST-positive, and 58.7% FAST-positive. If only FAST were used, 41.3% (*n* = 19) of posterior circulation infarcts would have been missed. The sensitivity and specificity of FAST and BE-FAST for PCS are shown in Table 2. BE-FAST had a higher AUC-ROC of 0.63 (95% CI 0.54–0.72) compared to FAST at 0.54 (95% CI 0.44–0.63) (Figure 1b).

Among anterior circulation strokes, BE-FAST was positive in 99.2%, while FAST was positive in 97.5%. FAST and BE-FAST sensitivity and specificity for anterior circulation ischaemic stroke are shown in Table 2. FAST had an AUC-ROC of 0.77 (95% CI 0.68–0.78), which was higher than BE-FAST at 0.73 (95% CI 0.72–0.81) (Figure 1c).

### 3.3. Performance of CT Perfusion

The performance of CTP in diagnosing AIS is summarised in Table 3. Of the posterior circulation infarcts, 56.5% (*n* = 26) were CTP negative. Of these, 25 were BE-FAST-positive and 14 were FAST-positive. The sensitivity of automated CTP for posterior circulation strokes was 42.2 (95% CI 27.7–56.8) and the specificity was 98.4 (95% CI 96.5–99.4). The AUC-ROC was 0.70 (95% CI 0.61–0.80).

A CTP was negative in 22.9% (*n* = 27) of anterior circulation infarcts. Of these, 27 were BE-FAST-positive and 26 were FAST-positive. The sensitivity of CTP for anterior circulation infarcts was 76.3 (95% CI 67.4–83.4), and the specificity was 98.4 (95% CI 96.5–99.4). The AUC-ROC was 0.88 (95% CI 0.83–0.92).

## 4. Discussion

Early recognition of stroke remains one of the key barriers in ensuring access to timely reperfusion therapy, and therefore, better functional outcomes. Although the time window for benefit has extended for both intravenous thrombolysis and EVT since the original pivotal trials, less than 10% of acute stroke patients undergo hyperacute therapies within the thrombolytic time window [30]. This rate is even less for rural and remote regions [31]. There is a greater risk of a worse functional outcome in a patient who has diminished or failing collateral circulation to an area affected by an acute ischaemic stroke and/or if the underlying brain tissue is severely hypoperfused [32]. Hence, the phrase ‘time is brain’ is still highly relevant. Posterior circulation strokes present a greater challenge, requiring implementation of better tools for recognition. Stroke detection scales such as FAST favour identification of symptoms more commonly associated with anterior circulation stroke. In our study, BE-FAST was a sensitive tool in identifying posterior circulation infarction compared to FAST, but suffered from poor specificity. CT perfusion improved the specificity of PCS detection but was less sensitive than BE-FAST. While prospective studies are necessary to validate our findings, the implementation of BE-FAST may improve the detection of posterior circulation infarction, ultimately resulting in earlier diagnosis, treatment, and better patient outcomes.

Several studies have previously assessed the utility of BE-FAST, although not in the context of MRI-confirmed posterior circulation infarction. In a retrospective analysis, Aroor et al. demonstrated a reduction in the number of strokes missed by FAST from 14.1% to 4.4% through the addition of balance and visual symptoms [17]. A subsequent prospective analysis by Pickham et al. in a prehospital setting found that the two scales had a comparable AUC [33]. In their study, the sensitivity of BE-FAST was 91% with a specificity of 56%, while FAST had a lower sensitivity at 76% but a higher specificity at 68%. Most recently, El Ammar et al. compared the use of BE-FAST in inpatient and community-onset stroke alerts [34]. Sensitivity among inpatient stroke alerts was 85%, with a specificity of 43%, while in community-onset cases, sensitivity was 94% with a specificity of 23%. These data, among others, were featured in a recent meta-analysis, which found BE-FAST to have a sensitivity of 68% with a specificity of 85%, and FAST with a sensitivity of 77% with a specificity of 60% [29].

Our analysis of BE-FAST and FAST generally revealed a higher sensitivity but lower specificity compared to previous studies. For posterior strokes, FAST missed 41.3% of cases, similar to previous reports [18]. Gulli et al. previously showed that the addition of ataxia or visual symptoms increased the sensitivity of FAST to 79.6% and 81.5%, respectively [18], although they did not analyse the inclusion of both. While we are unable to compare the performance of BE-FAST in anterior and posterior circulation strokes to other studies, our data had a trend of a high sensitivity and low specificity. We suspect this may be in part due to the differences in training and experience of administration of the BE-FAST scale. At our institution, most stroke codes are activated by the ambulance or triage personnel, prior to assessment by medical personnel. Our study relied only upon these initial assessments. This most likely resulted in a large number of false positive stroke alerts but also a high negative predictive value for PCS detection. Regardless, our study reflects the experience at a tertiary centre with a comprehensive stroke service.

We also examined the sensitivity of automated CTP in identifying AIS. A recent meta-analysis demonstrated that automated CTP had a pooled sensitivity of 82% with a specificity of 96% in the diagnosis of AIS [35]. For posterior circulation infarcts, the sensitivity has been found to range from 31% to 77% [23,36] with a high specificity of >90% [36]. The sensitivity and specificity of CTP is known to vary according to stroke location [23,24]. Our sensitivity at 66.7% for the detection of all AIS was lower than other studies. For PCS, Sporns et al. found a higher sensitivity of CTA (44% versus 17%) and CTP (77% versus 42%) compared to our study [23]. This may relate to differences in time from symptom onset to CT, inclusion criteria, stroke location, vessel involvement, and methodology, for example, the evaluation of images by experienced neuroradiologists as opposed to the use of automated CTP [23]. It has been recently demonstrated that commonly used perfusion thresholds in CTP may be inadequate in the characterisation of posterior circulation infarcts, and more optimal thresholds have been proposed that may increase the sensitivity of automated CTP [24].

Previous studies have demonstrated an improved detection of posterior circulation strokes when multimodal CT analysis is used compared to non-contrast CT and automated CTP alone [36]. Similarly, the expert assessment of raw CTP maps yield a lower false negative rate compared to an automated software-based detection of posterior circulation infarcts [37]. Expert examination of raw perfusion maps is therefore encouraged to avoid missing lesions. However, this relies on timely access to expert neuroradiology services. This is also the case for the accurate diagnosis of LVOs using CTA. Even when evaluated by radiologists, LVOs in CTA have been missed in 7–20% of cases [38,39]. In one study, this increased to 38% in arterial occlusions of smaller arteries [39]. A combination of CTP with CTA has been demonstrated to increase the sensitivity and specificity of LVO detection [40]. There are also efforts to automate LVO detection using algorithms and machine learning [41,42]. While these are being developed, the use of automated CTP remains a means to provide equity to all centres, in particular those that do not have 24 h access to specialist radiology services. The utility of CTP in the detection of PCS is an area of interest that warrants further research.

In our study, BE-FAST outperformed CTP in identifying posterior circulation infarcts. CT perfusion, however, benefited from a higher specificity compared to BE-FAST. Given this, a combination of BE-FAST and CTP may be used to improve the detection of posterior circulation infarcts. Using Fagan’s nomogram, assuming a pre-test probability of 8.3%—the percentage of stroke codes identified as posterior circulation infarcts in our sample—a negative BE-FAST would yield a post-test probability of 2.0%. On the other hand, given its limited specificity, a positive test would yield a post-test probability of 9.0%. If followed by CTP, however, a positive result would yield a post-test probability of 75.9%, while negative neuroimaging would yield a probability of 5.6%. As patients do not necessarily present with a known vascular distribution, more sensitive and specific tools are still necessary to improve stroke detection. Nonetheless, a combination of BE-FAST and CTP could be a useful method to identify posterior circulation infarcts, until more sensitive and specific tools become available.

Another issue for newer stroke detection tools such as BE-FAST is their impact on resource utilisation. In the present study, if FAST was the tool being used for stroke code activation rather than BE-FAST, there would have been 137 less stroke notifications during the four-month time period of the study equating to a 25% reduction in acute stroke service workload. The trade-off for the identification of 18 more cases of PCS by using BE-FAST was the need to review an extra 117 cases who were stroke mimics. Education sessions for paramedic and ED staff in anomalies of eyes and balance may improve the specificity of BE-FAST for PCS detection. However, facial droop, arm weakness, and speech disturbance are still the most easily recognisable symptoms and signs that distinguish stroke from non-stroke patients. The utility of education to ED medical and nursing staff to distinguish PCS from non-stroke cases at the bedside as opposed to the activation of a stroke code call for all BE-FAST-positive presentations is unclear.

This study had several limitations. First, the retrospective analysis prevented the investigation of whether earlier recognition led to faster treatment and better outcomes. Our analysis also focused on a single centre, and is biased by the protocols within this centre. Therefore, multi-centre prospective studies are necessary to validate our findings. Additionally, only patients who received a stroke code were included in the study. This causes a bias toward the sensitivity analysis as patients whose symptoms did not result in a stroke code or who had negative MRI findings [43,44] but may have had a stroke were not included. Haemorrhagic infarcts were also excluded from analysis, limiting the external validity of the study. Finally, we only relied upon automated CTP rather than perfusion maps interpreted by an experienced neuroradiologist, which may have altered the performance of CTP in our analysis. Similarly, we relied upon the reports generated by local radiologists for the evaluation of CT and CTA, which may account for the lower frequency of abnormal findings compared to other studies. This, however, again reflects a real-life clinical scenario wherein expert neuroradiology services are not available.

### Future Directions

Future directions include artificial intelligence and machine learning for CTP to improve its sensitivity and optimise the detection of both anterior and posterior circulation ischaemic stroke [45,46]. Stroke scales have more recently focused on the identification of LVO stroke amenable to EVT more so than the differentiation of anterior and PCS, but there are several screening tools such as Recognition of Stroke in the Emergency Room (ROSIER) that may be useful in distinguishing stroke from non-stroke presentations [47]. ROSIER encompasses additional findings to face, arm, and leg weaknesses, including altered consciousness and visual field defect that may have moderate to high sensitivity and specificity for anterior and PCS detection when applied in the prehospital and ED setting [47]. Further studies are necessary to examine its utility in PCS. Furthermore, other scoring systems have also been proposed specifically for PCS, including the Israeli Vertebrobasilar Stroke Scale [48] and Adam’s Scale of Posterior Stroke [49]; however, these require further clinical validation and comparison to existing tools.

Telestroke medicine services operating through remote communication with expert health care providers may also improve the accuracy of stroke diagnosis and differentiation of LVO from non-LVO stroke. Telestroke medicine may be useful in rural, remote, and metropolitan hospital settings and obviate the need for over-reliance on stroke screening tools [50].

Finally, video head-impulse testing (vHIT) may be helpful in the differentiation of acute dizziness due to vestibular neuritis as opposed to posterior circulation stroke. A prospective analysis has demonstrated that abnormal vHIT results are rare in patients with acute PCS [51], while a recent review of the literature has advocated for further research to validate the clinical adoption of vHIT [52].

## 5. Conclusions

In conclusion, our study found that the incorporation of balance and eye assessment into the FAST protocol improved PCS detection, although yielded more false positives. Although highly specific, automated CT perfusion was less sensitive than BE-FAST in the detection of PCS.

## Figures and Tables

**Figure 1 jcm-13-05912-f001:**
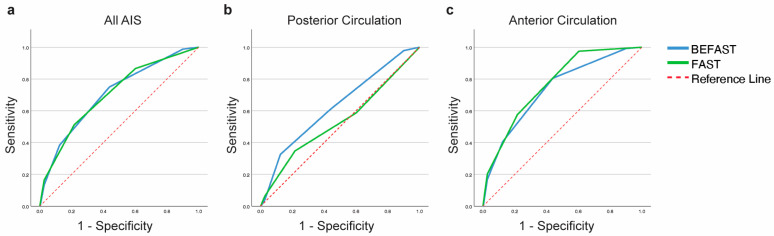
BE-FAST and FAST receiver operating characteristic curves for (**a**) all ischaemic strokes, (**b**) posterior circulation strokes, and (**c**) anterior circulation strokes. AIS: acute ischaemic stroke.

**Table 1 jcm-13-05912-t001:** Patient demographics in all stroke codes.

Demographics	Not AIS	Anterior Circulation	Posterior Circulation	*p* Value ^a^
*n* (%)	392 (70.5)	118 (21.2)	46 (8.3)	
Median age	65.0 (51.0–77.0)	74.0 (64.0–81.0)	70.5 (61.5–83.0)	0.725
Sex *M/F* (%)	204/188 (52.0/48.0)	66/52 (55.9/44.1)	30/16 (65.2/34.8)	0.278
Median ABCD2 score	4.0 (3.0–5.0)	5.0 (5.0–6.0)	5.0 (3.75–5)	< 0.001
Median NIHSS	1.0 (0.0–4.0)	8.0 (2.0–15.0)	4.0 (2.0–8.5)	0.018
AF prevalence *n* (%)	41 (10.5)	42 (35.6)	13 (69.6)	0.380
BE-FAST-positive *n* (%)	353 (90.1)	117 (99.2)	45 (97.8)	0.487
FAST-positive *n* (%)	236 (60.2)	115 (97.5)	27 (58.7)	< 0.001
CTP positive *n* (%)	5 (1.3)	87 (73.7)	19 (41.3)	< 0.001
Therapy n (%)				
EVT	0 (0)	25 (21.2)	5 (10.9)	0.125
IVT	6 (1.5)	11 (9.3)	10 (21.7)	0.033
EVT + IVT	0 (0)	14 (11.9)	1 (2.2)	0.053

Values are presented as median (interquartile range). ^a^ Comparing anterior and posterior circulation demographics; ABCD2: Age, Blood Pressure, Clinical features, Duration (of transient ischaemic attack), Diabetes Mellitus; AF: atrial fibrillation; BE-FAST: Balance Eyes—Face, Arms, Speech, Time; CTP: computerised tomography perfusion; EVT: endovascular thrombectomy; FAST: Face, Arms, Speech, Time; IVT: intravenous thrombolysis; NIHSS: National Institutes of Health Stroke Scale.

**Table 2 jcm-13-05912-t002:** Performance of BE-FAST and FAST in the diagnosis of acute ischaemic stroke.

		BE-FAST % (95% CI)	FAST % (95% CI)
All AIS	Sensitivity	98.8 (95.7–99.9)	86.6 (80.4–91.4)
	Specificity	10.0 (7.2–13.4)	39.8 (34.9–44.8)
	PPV	31.5 (30.7–32.3)	37.6 (35.2–40.0)
	NPV	95.1 (82.7–98.8)	87.6 (82.5–91.4)
	AUC-ROC	0.71 (0.66–0.75)	0.70 (0.66–0.75)
Posterior Circulation	Sensitivity	97.8 (88.5–99.9)	58.7 (43.2–73.0)
	Specificity	10.0 (7.2–13.4)	39.8 (34.9–44.8)
	PPV	11.3 (10.8–11.9)	10.3 (8.1–12.9)
	NPV	97.5 (84.6–99.6)	89.1 (85.1–92.2)
	AUC-ROC	0.63 (0.54–0.72)	0.54 (0.44–0.63)
Anterior Circulation	Sensitivity	99.2 (95.4–100.0)	97.5 (92.8–99.5)
	Specificity	10.0 (7.2–13.4)	39.8 (34.9–44.8)
	PPV	23.1 (19.5–27.1)	32.8 (30.9–34.7)
	NPV	97.5 (84.4–99.7)	98.1 (94.4–99.4)
	AUC-ROC	0.73 (0.72–0.81)	0.77 (0.68–0.78)

AIS: acute ischaemic stroke; AUC-ROC: area under the receiver operating characteristic curve; BE-FAST: Balance and Eyes–Face, Arms, Speech, Time; CI: confidence interval; FAST: Face, Arms, Speech, Time; NPV: negative predictive value; and PPV: positive predictive value.

**Table 3 jcm-13-05912-t003:** Performance of automated CTP in diagnosis of AIS.

	Parameter	Value
All AIS	Sensitivity	66.7 (58.8–73.9)
	Specificity	98.7 (96.9–99.6)
	PPV	95.5 (89.8–98.1)
	NPV	87.5 (84.9–89.7)
	AUC-ROC	0.83 (0.78–0.87)
Posterior Circulation	Sensitivity	42.2 (27.7–56.8)
	Specificity	98.4 (96.5–99.4)
	PPV	76.0 (57.2–88.3)
	NPV	93.4 (91.7–94.8)
	AUC-ROC	0.70 (0.61–0.80)
Anterior Circulation	Sensitivity	76.3 (67.4–83.8)
	Specificity	98.4 (96.5–99.4)
	PPV	93.6 (86.7–97.0)
	NPV	93.2 (90.7–95.0)
	AUC-ROC	0.88 (0.83–0.92)

Values presented as % (95% confidence intervals). AIS: Acute ischaemic stroke; AUC-ROC: area under the receiver operating characteristic curve; CTP: computerised tomography perfusion; NPV: negative predictive value; and PPV: positive predictive value.

## Data Availability

The data that support the findings of this study are not publicly available due to the information contained, which could compromise the privacy of the research participants, but are available from the corresponding author upon reasonable request.

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
