# Peer review of "Enhancing Stroke Recognition: A Comparative Analysis of Balance and Eyes–Face, Arms, Speech, Time (BE-FAST) and Face, Arms, Speech, Time (FAST) in Identifying Posterior Circulation Strokes"

_jcm, 2024, doi:10.3390/jcm13195912_

Round 1

Reviewer 1 Report (Previous Reviewer 1)

Comments and Suggestions for Authors

Recommendation: Accept.

Comments on the Quality of English Language

N/A

Author Response

We thank the reviewer for their recommendation to accept the manuscript.

Reviewer 2 Report (Previous Reviewer 2)

Comments and Suggestions for Authors

Dear authors,

I understand that you mainly compared different neurological examinations in acute ischaemic stroke and found BEFAST superior to FAST, especially in the posterior circulation. Obviously you did not carefully analyse the imaging, but maybe relied on routine reports and the decision of a perfusion software. This should be stated in methods, results and discussion.

It should be explained why you found in the posterior circulation only a detection rate of 6,5 % by plain CT, 17% by CT-A, and 41 % (i.e. 59% failure) by P-CT when in ref.  [21] you cited, it was 21% (>triple), 44% (>x2.5), and 77% (i.e. 23% (<x2.5) failure) respectively.

In line 86 are quite wrong statements. You say:

Computerised Tomography (CT) brain imaging including angiography (CTA) is commonly normal in the hyperacute setting.

It is right that brain hypodensity seen on plain CT takes often several hours until infarction develops.

But an hyperdense vessel sign in thin slice CT, CT-A and P-CT are immediately positive, within the second, when the occlusion occurs.

Author Response

Comment 1: I understand that you mainly compared different neurological examinations in acute ischaemic stroke and found BEFAST superior to FAST, especially in the posterior circulation. Obviously you did not carefully analyse the imaging, but maybe relied on routine reports and the decision of a perfusion software. This should be stated in methods, results and discussion.

Response 1: We thank the reviewer for their comment. We have clarified throughout the manuscript that we relied upon local radiologist reports and automated CTP. We acknowledge this as a limitation, however also posit that our study reflects what is often encountered in most clinical settings, especially when expert neuroradiology services are not routinely available.

Comment 2: It should be explained why you found in the posterior circulation only a detection rate of 6,5 % by plain CT, 17% by CT-A, and 41 % (i.e. 59% failure) by P-CT when in ref.  [21] you cited, it was 21% (>triple), 44% (>x2.5), and 77% (i.e. 23% (<x2.5) failure) respectively.

Response 2: We thank the reviewer for highlighting this. We have expanded the discussion and limitations to suggest some reasons as to why our findings are lower than previously reported. One reason of course again is a lack of expert neuroradiology assessment. In the paper we cite with the higher statistics (Sporns et al, ref 23), expert neuroradiologists evaluated the imaging, and if there were any discrepancies, the imaging were re-evaluated to reach consensus. It is also likely that our sample had a higher rate of brainstem and small perforator involvement. Sporns et al acknowledge that their centre was a tertiary referral hospital for basilar occlusion, and therefore may have had a disproportionately high number of large vessel occlusion cases. Finally the study by Spprn et al had inclusion criteria of patients clinically suspected of having PCS whereas in our study we assessed FAST or BEFAST positive ED stroke calls.

Comment 3: In line 86 are quite wrong statements. You say:

Computerised Tomography (CT) brain imaging including angiography (CTA) is commonly normal in the hyperacute setting.

It is right that brain hypodensity seen on plain CT takes often several hours until infarction develops.

But an hyperdense vessel sign in thin slice CT, CT-A and P-CT are immediately positive, within the second, when the occlusion occurs.

Response 3: We thank the reviewer for their comment. We have altered this sentence to reflect the utility of CT abnormalities in the first instance, including the hyper dense vessel sign.

Round 2

Reviewer 2 Report (Previous Reviewer 2)

Comments and Suggestions for Authors

Dear authors, you improved the article essentially. Thank you.

This manuscript is a resubmission of an earlier submission. The following is a list of the peer review reports and author responses from that submission.

Round 1

Reviewer 1 Report

Comments and Suggestions for Authors

Review Summary

This paper evaluates the effectiveness of BEFAST compared to FAST in identifying posterior circulation strokes (PCS) using retrospective data from a comprehensive stroke center. It highlights the importance of improving stroke detection methods, especially for posterior events which present unique diagnostic challenges.

Current Advances

Recent advancements in stroke recognition have focused on enhancing traditional mnemonics like FAST to include additional symptoms indicative of posterior circulation strokes, leading to the development of BEFAST. This paper contributes to the ongoing debate by providing empirical evidence comparing the sensitivity and specificity of these two mnemonics in a clinical setting.

Review Comments

1. Methodological Rigor:

The retrospective design is appropriate for this type of study, though a prospective approach would strengthen the evidence. The inclusion criteria are clearly defined, but the study would benefit from a more detailed explanation of how data were handled to minimize bias.

2. Clinical Implications:

The discussion on clinical implications is strong, emphasizing the potential for BEFAST to improve PCS detection. Further elaboration on how these findings could be integrated into current stroke protocols and their potential impact on patient outcomes would be beneficial.

3. Limitations and Future Research:

The paper acknowledges key limitations, such as the retrospective design and the focus on a single center. Future research should include multi-center prospective studies to validate these findings. Exploring the integration of automated imaging techniques with BEFAST could also be a promising avenue.

4. Writing and Clarity:

The manuscript is well-written, with clear and concise language. Minor grammatical corrections and improved flow in the discussion section would enhance readability.

5. Comparison with Existing Literature:

We found a novel paper addressing the same topic with this paper, please compare the differences.

Hogge C, Goldstein LB, Aroor SR. Mnemonic utilization in stroke education: FAST and BEFAST adoption by certified comprehensive stroke centers. Front Neurol. 2024;15:1359131. Published 2024 Mar 12. doi:10.3389/fneur.2024.1359131

Overall Recommendation

Minor Revision: The paper is of high quality and makes a significant contribution to the field. Addressing the minor methodological and presentation improvements suggested above will further enhance its impact and clarity.

Comments on the Quality of English Language

Minor check.

Reviewer 2 Report

Comments and Suggestions for Authors

The comparison of FAST and BEFAST as clinical exams is useful. But to compare with "automated" C-TP alone, what ever that means, does not make any sense. If CT with contrast in stroke is done, CT-A and CT-P should be seen by a neuroradiologist and then it is impossible only to achieve a 42 % sensitivity. For example in a Percheron artery occlusion by a P1-embolus, it has to be analysed by a physician, not left by an automate for perfusion alone. The images are there to be seen!

Maybe for the paper, it would be better to leave out the CT-P results. Else you should report, which parts of the brain were comprised by the CT-P and if CT-A was calculated (which can be done from CT-P data ore by separate injection) and why CT-A is not part of the evaluation. Why should you look only at a part of the CT-images?